# Healthcare Workers’ Willingness to Receive Influenza Vaccination in the Context of the COVID-19 Pandemic: A Survey in Southern Italy

**DOI:** 10.3390/vaccines9070766

**Published:** 2021-07-09

**Authors:** Gabriella Di Giuseppe, Concetta P. Pelullo, Andrea Paolantonio, Giorgia Della Polla, Maria Pavia

**Affiliations:** 1Department of Experimental Medicine, University of Campania “Luigi Vanvitelli”, Via Armanni, 5, 80138 Naples, Italy; gabriella.digiuseppe@unicampania.it (G.D.G.); concettapaola.pelullo@unicampania.it (C.P.P.); andrea.paolantonio@studenti.unicampania.it (A.P.); 2Health Direction, Teaching Hospital of the University of Campania “Luigi Vanvitelli”, Via S. Maria di Costantinopoli 104, 80138 Naples, Italy; giorgia.dellapolla@unicampania.it

**Keywords:** COVID-19 pandemic, healthcare workers, influenza vaccination, Italy, survey, vaccination coverage

## Abstract

This cross-sectional survey was designed to evaluate hospital healthcare workers’ (HCWs) willingness to receive the influenza vaccination during the COVID-19 pandemic and to identify the related determinants, since it is plausible that the two epidemics will coexist in future winters. Overall, 68% out of 490 participants expressed their willingness to receive influenza vaccination in the 2020/21 season, with 95% of those ever and 45.8% of those never vaccinated in the previous six influenza seasons. Belief that influenza vaccine is useful in distinguishing influenza symptoms from those of COVID-19 and that the influenza vaccine is useful to prevent influenza in hospital settings, willingness to receive COVID-19 vaccination, having no concern about influenza vaccine side effects, concern about the possibility to transmit influenza to hospitalized patients, and influenza vaccination in previous years were all predictors of willingness to receive influenza vaccination. In the context of the COVID-19 pandemic, a relevant increase in the willingness to undergo influenza vaccination was reported. Therefore, interventions focused primarily on enabling factors are needed to promote the adherence to influenza vaccination in future seasons among HCWs.

## 1. Introduction

Influenza is known to be a global public health priority [1], affecting about 5–10% of the world population each year, with an estimated number of deaths ranging from 250,000 to 500,000 worldwide [2].

The transmission of influenza within healthcare facilities is widely reported in the literature, and healthcare workers (HCWs) represent a priority group for seasonal influenza vaccination recommendations. Immunization against influenza not only reduces the risk of infection among HCWs and the potential consequent defection and disruption of healthcare services, but also improves patient safety, reducing morbidity and mortality among the most vulnerable subjects [3].

In Italy, influenza affects almost 9% of the Italian population every year and, in particular, during the last season, there were 2,400,000 cases. Data on influenza are provided through InfluNet, a surveillance system coordinated by the National Institute of Health (NIH). It is based on the participation of general practitioners and pediatricians who report the number of cases of influenza like illness (ILI) observed each week among their patients [4]. Moreover, every year the Ministry of Health issues a document containing information on influenza epidemiological and virological surveillance, and providing recommendations for the seasonal influenza vaccination campaign. In this document, the target groups for influenza vaccination, such as HCWs, are identified and thresholds for monitoring the success of the campaigns are defined [5]. Although influenza vaccination in HCWs is strongly recommended, adherence is still very poor [6,7,8,9,10]. Accurate information on determinants of “vaccine hesitancy” among HCWs is not available, although it has been reported that it is mainly related to vaccine safety, especially in the case of the influenza vaccine [11].

In the context of poor adherence to influenza vaccination among HCWs, the co-circulation of influenza and SARS-CoV-2 viruses in the autumn/winter season 2020/21 has represented a public health challenge. It has been reported that coinfections with SARS-CoV-2 and influenza or other respiratory viruses might produce more severe diseases [12,13], and differential diagnosis between the two syndromes might be facilitated if HCWs were vaccinated against influenza.

Thus, understanding HCWs willingness to receive influenza vaccination also has relevant implications for discerning whether the pandemic may influence HCWs willingness and related behavior concerning influenza vaccination.

Therefore, this cross-sectional survey was designed to evaluate HCWs’ willingness to receive influenza vaccination during the COVID-19 pandemic and to identify related determinants in Italy. This knowledge may inform interventions for the upcoming influenza seasons, since it is plausible that the two epidemics will coexist in the future winters.

## 2. Materials and Methods

### 2.1. Study Design and Setting

This cross-sectional study was carried out in December 2020 and took place in three randomly selected public general hospitals distributed across Southern Italy (Campania region). The study population of HCWs was randomly sampled from the selected hospitals. The sample size was calculated by using the single population proportion formula with the assumption that 20% [10,14] of HCWs would be willing to undergo influenza vaccination, a confidence level of 95%, a margin of error at 5%, and considering a response rate of 60% [15,16,17]. Therefore, the final “minimum” needed sample size was 410 HCWs. Ethical approval was obtained from the Ethics Committee of the Teaching Hospital of the University of Campania “Luigi Vanvitelli”.

### 2.2. Data Collection

Before data collection, the hospital directors received an information letter explaining the objectives and the methodology of the study, in order to obtain their approval. After the approval was received, the questionnaires were distributed to the selected HCWs with a cover letter that explained the purpose of the study, that participation was voluntary, and that confidentiality would be granted. The anonymization of any personal identifiers was assured, and an informed consent form to be signed by those who accepted to participate was also included. Moreover, to ensure maximum recruitment of the HCWs and to manage non-responders, follow-up visits were scheduled twice a week to each hospital. There were no incentives offered to HCWs who accepted to participate in the survey.

### 2.3. Survey Questionnaire

The questionnaire was built ad hoc for this survey after an extensive review of the literature assessing attitudes and adherence to vaccinations among HCWs [10,18,19,20,21]. The structured questionnaire was self-administered and consenting participants were asked to answer questions on four themes: (1) socio-demographic and professional characteristics (gender, age, nationality, marital status, parenthood, year of degree, professional role, ward, years in practice, average working hours and, average number of patients followed per week); (2) attitudes about influenza and COVID-19, and related vaccination strategies (perception of the risk of the disease, of safety and effectiveness of preventive measures, etc.); (3) behaviors regarding influenza vaccination uptake in the preceding influenza seasons (years 2014/15–2019/20); (4) sources of information, their quality and the need for additional information on influenza vaccination. Beliefs about influenza and related vaccination were measured on a 3-point Likert-type scale with options for agree, uncertain, and disagree, which, for the purpose of analysis, have been dichotomized as 1 for “agree” and 0 for “uncertain” and “disagree”, whereas questions regarding the perception of the risk of contracting influenza and COVID-19 in the workplace, of transmitting influenza to patients, and beliefs about the usefulness and the safety of the influenza and the COVID-19 vaccine were measured on a 10-point Likert scale ranging from 1 (not worried) to 10 (extremely worried). Willingness to undergo influenza vaccination in the 2020/21 season and influenza vaccination uptake in the 2014/15–2019/20 seasons were investigated through response options including ‘‘yes” and ‘‘no”. For influenza vaccination uptake the option “do not remember” was also available. Finally, a list of potential reasons for having undergone influenza vaccination or not during the previous seasons was proposed with “yes” or “no” options for response for one or more reasons. The study was preceded by a pilot test among 50 HCWs, to evaluate the readability, clarity and correct sequence of the items. After conducting the pilot study, the reliability of the questionnaire was evaluated through the Cronbach’s alpha, which indicated a high internal consistency. Calculation of the content validity indicated the unanimous agreement with the content and clarity of the questions (Cronbach’s alpha = 0.73).

### 2.4. Statistical Analysis

Descriptive and inferential statistics were used for the analyses of the data. First, a descriptive analysis was conducted to explore the main characteristics of the sample; second, the univariate analysis was performed using the chi-squared test for the categorical variables and the Student’s *t*-test for the continuous variables. Then, multivariate logistic models were constructed to identify factors associated with the following outcomes of interest: willingness to receive influenza vaccination (no = 0; yes = 1) (Model 1); willingness to receive influenza vaccination among those who had not received vaccination influenza in the previous six years vs. all others (no = 0; yes = 1) (Model 2). The independent variables that were shown to be associated at the univariate analysis or that were judged to potentially have influence on the investigated outcomes were included in the appropriate model.

The following independent variables were included in all models: gender (male = 0; female = 1); professional role (physician = 1, nurse = 2, other = 3); usefulness of influenza vaccine to distinguish influenza symptoms from those of COVID-19 (no = 0; yes = 1); willingness to receive COVID-19 vaccine (no = 0; yes = 1); concern about influenza vaccine side effects (continuous); concern about the risk of transmitting influenza to hospitalized patients (continuous); belief that influenza vaccine is useful to prevent influenza in hospital settings (continuous). In Model 1, the independent variable having been vaccinated against influenza in the previous years (no = 0; yes = 1) was also included.

Odds ratios (ORs) and 95% confidence intervals (CIs) were presented in the logistic regression models. All statistical tests were two-sided, and the level of statistical significance was set at 0.05. Analyses were performed with the Stata software, version 15 [22].

## 3. Results

Of the 700 HCWs invited to participate, 490 agreed and returned the survey for an overall response rate of 70%. Table 1 shows the main investigated characteristics of the participating HCWs. The mean age was 50.7 years (SD = 10.5); 54.4% were females, 74.8% were married/cohabitant; 72.6% had children; slightly more than half (52%) were nurses; 34.1% worked in medical wards; the mean time from degree was 25.8 years (SD = 10.8); and the mean number of hours worked per week was 36.5 (SD = 4.4).

The vast majority (81%) agreed that HCWs can be a source for influenza infection for patients; 69.2% considered HCWs to be at high risk of contracting influenza; 66.3% considered influenza preventable and only 36.7% considered it a serious disease. Overall, 78% and 76.1% declared themselves to be favorable to recommended vaccinations and in particular to influenza vaccination for HCWs, respectively, whereas only 33.5% were favorable to mandatory influenza vaccination for HCWs.

The uptake of influenza vaccine in the previous influenza seasons showed an increase in the temporal trend ranging from 17.3% in the 2014/15 season to 40.8% in the 2019/20 season, with 273 (55.7%) and 75 (15.3%) reporting to have never and always undergone influenza vaccinations in the 2014/15–2019/20 seasons, respectively. Willingness to receive influenza vaccination in the 2020/21 season was expressed by 333 (68%) participants and by 95.8% of those who had ever and by 45.8% of those who had never undergone influenza vaccination in the previous six seasons. Among those who expressed willingness to receive it, the most commonly reported reasons were the reduction in risk of infection (74.9%), the effectiveness (55.1%) and safety (53.3%) of the vaccine, and that they believed themselves at risk of contracting influenza (49.2%), whereas among those who were not willing to undergo influenza vaccination, the main reasons were concerns about the usefulness of the vaccine (38.6%), considering themselves not to be at risk of contracting influenza (34.3%), fear of adverse events (25.7%), and doubts about the effectiveness of the vaccine (15%) (Table 2).

Almost all HCWs stated to have obtained information about influenza vaccination (96.1%). The preferred sources of information were scientific journals (56.5%), mass media (41.4%), the Internet (25.3%), and colleagues (20.4%). Moreover, 53.9% reported the need for additional information about influenza vaccination.

Table 1 reports also the results of the univariate analysis on willingness to receive influenza vaccination according to several characteristics. It shows that this willingness was significantly associated with being male (74.7% vs. 62.5%); being physicians (84.3%) compared to nurses (61.6%) or other HCWs (62.4%); considering influenza a serious (82.7% vs. 59.3%) and preventable disease (75.1% vs. 53.9%); considering HCWs a source of influenza infection for patients (70.5% vs. 57%); having a positive attitude towards influenza vaccination in HCWs (79.4% vs. 31.6%); considering HCWs at higher risk of contracting influenza (75.8% vs. 50.3%); being favorable to mandatory influenza vaccination in HCWs (84.8% vs. 59.5%); considering influenza vaccination useful to distinguish influenza symptoms from COVID-19 symptoms (81.4% vs. 53%); willingness to receive COVID-19 vaccination (78.5% vs. 34.7%); being concerned about contracting influenza in the workplace (6.35 ± 2.57 vs. 5.55 ± 2.63); being concerned about the possibility of transmitting influenza to hospitalized patients (6.32 ± 2.45 vs. 5.01 ± 2.6); believing that influenza vaccine is useful to prevent influenza in hospital settings (7.96 ± 1.97 vs. 5.92 ± 2.40); not being concerned about influenza vaccine side effects (5.20 ± 3.03 vs. 6.64 ± 2.35); having received influenza vaccination in the previous influenza season (99% vs. 46.5%); having used scientific journals as source of information on influenza vaccination (75.8% vs. 57.8%); and perceiving the quality of information received on influenza vaccination as good/very good/excellent (71.9% vs. 56.7%) (Table 1).

Many of these predictors were confirmed in the multivariate analysis, specifically: considering the influenza vaccination a useful tool to differentiate symptoms of influenza from those of COVID-19 (OR = 2.01; 95% CI = 1.23–3.57); belief that influenza vaccine is useful to prevent influenza in hospital settings (OR = 1.25; 95%CI = 1.1–1.42); willingness to receive COVID-19 vaccination (OR = 3.41; 95% CI = 1.87–6.22); not being concerned about influenza vaccine side effects (OR = 0.82; 95% CI = 0.73–0.91); being concerned about the risk of transmitting influenza to hospitalized patients (OR = 1.15; 95% CI = 1.03–1.29); and having been vaccinated against influenza in the previous years (OR = 2.78; 95% CI = 1.85–4.19) were all significantly associated with willingness to receive influenza vaccination (Model 1 in Table 3). Moreover, considering the influenza vaccination a useful tool to differentiate symptoms of influenza from those of COVID-19 (OR = 1.65; 95% CI = 1.03–2.63), being female (OR = 1.61; 95%CI = 1.03–2.5), and not being concerned about influenza vaccine side effects (OR = 0.87; 95% CI = 0.8–0.95) were all significantly associated with willingness to receive influenza vaccination among those who had never undergone influenza vaccination in the preceding six influenza seasons, and willingness to receive the COVID-19 vaccination almost resembled association (*p* = 0.094) (Model 2 in Table 3).

## 4. Discussion

Within the large body of literature exploring the propensity of HCWs to receive seasonal influenza vaccination, this study has investigated this willingness in the context of the ongoing COVID-19 pandemic, exploring the intention to uptake seasonal influenza vaccination and the related determinants, as well as the determinants of change in the willingness to be vaccinated in those who had not undergone this vaccination in the preceding six influenza seasons. In particular, the hypothesis of the study was that HCWs would be more willing to undergo influenza vaccination to diminish their own risk and that of their patients of coinfection with COVID-19 and to reduce the problems related to differential diagnosis between the two diseases. The findings of the study may be of interest to suggest the most appropriate interventions aimed at increasing seasonal influenza vaccination coverage in this strategic population group, by the identification of cues of action for the future influenza vaccination strategies, considering that it is plausible that the two epidemics will continue to coexist, at least in the near future.

The overall willingness to receive influenza vaccination reported by the participants was 68%, a value that is within those reported in the systematic review by Bish and colleagues (22% to 83%) [23], and slightly higher compared to the mean adherence to influenza vaccination, which is generally less than 30% in HCWs, as reported by Dini and colleagues [7]. In the context of the COVID-19 pandemic, however, it is more interesting to investigate whether there has been a change in the intention to uptake the influenza vaccination compared to adherence in previous years. Indeed, consistently with the literature on this topic, one of the strongest predictors of this willingness was adherence to influenza vaccination in the previous seasons. It is of note that 95% of those who had undergone influenza vaccination in the previous seasons reported also being willing to undergo it in the present one. However, the most remarkable result is that almost half of those who reported themselves to have never had an influenza vaccination in the preceding six influenza seasons, were now willing to receive it. Since the proportion of HCWs who reported that they had never undergone this vaccination in the six preceding seasons was high (55.7%), this finding is stimulating and worthy of a detailed analysis. It should be noted that data have shown an increasing trend in the uptake of influenza vaccination in the preceding six seasons, ranging from 17.3% in the 2014/15 season to 40.8% in the 2019/20 season, suggesting that strategies to develop higher adherence to this vaccination in HCWs have become increasingly successful, although the uptake rates are still unsatisfactory in this target group. Therefore, this additional increase in the willingness may be partly the consequence of this favoring temporal trend; however, it is plausible that the context of the COVID-19 pandemic might have been responsible, at least in part, for this increased willingness. These conclusions have been drawn by other studies that have investigated the role of the COVID-19 pandemic in influenza vaccination from different perspectives. In a survey conducted on HCWs in the UK, 44% of the participants reported they were more likely to have an influenza vaccine in the 2020/21 season due to the COVID-19 pandemic [24], whereas in an Italian teaching hospital the increase in influenza vaccination uptake among HCWs increased from 24.19% in the 2019/20 season to 54.56% in the 2020/21 season, although a rise to 30.35% was predicted by a model based on data from four previous campaigns [25]. Moreover, the results of a web-based survey performed in university students showed that 77.5% of them were willing to receive the flu vaccine, and among the predictive factors to undertake flu vaccination, there was a high level of concern and perceived vulnerability to the COVID-19 pandemic [26]. Therefore, the awareness of the relevance of influenza vaccination in the context of SARS-CoV-2 circulation is widespread in several settings and populations, stimulating the implementation of strategies to improve influenza vaccination coverage rates.

Further suggestions on the most appropriate measures to put in place to promote HCWs influenza vaccination in the upcoming seasons may be drawn by the examination of the factors that predicted willingness to receive it. We have already mentioned the role of adherence to influenza vaccination in the previous seasons, which has been repeatedly reported in studies investigating predictors of willingness and adherence to influenza vaccination [21,27,28,29], demonstrating that the experience of vaccination is satisfactory for HCWs, and represents an incentive for future vaccinations. Since in Italy influenza vaccination is recommended, but not mandatory in HCWs, the positive role of adherence to previous influenza vaccination campaigns pertains to those who voluntarily underwent vaccination. Indeed, in this study, although 76.1% expressed to be favorable to influenza vaccination in HCWs, only one third would favor mandatory influenza vaccination. Although mandatory vaccination for children is a consolidated but always controversial strategy [30,31], the debate on the opportunity for mandatory influenza vaccinations in HCWs is also long-standing [32], and has been relaunched during the COVID-19 pandemic [33]. The negative attitude towards mandatory influenza vaccination expressed by the majority of the surveyed HCWs is higher than that which resulted from a recent meta-analysis investigating this issue, reporting a pooled estimate of 61% of HCWs that were in favor of this policy [33]. In the same meta-analysis, however, a great heterogeneity among studies was revealed, with the lowest acceptance of mandatory policies expressed by European HCWs [33].

Confidence in the usefulness and the safety of a vaccine are consolidated predictors of willingness and adherence to vaccinations, and this was also the case for the surveyed HCWs, since considering influenza vaccination as a useful and safe prevention tool in the hospital settings was positively associated with willingness to receive the vaccine. Moreover, concern about the risk of transmitting influenza to hospitalized patients may be interpreted as an indicator of HCWs’ awareness of their critical role in the protection of vulnerable subjects. All of these determinants have already been reported in the literature as determinants of a positive attitude of HCWs towards receiving influenza vaccination [7,34,35,36], whereas it is noteworthy that in this last season willingness was also influenced by the COVID-19 pandemic. This is suggested by the findings that considering the influenza vaccination a useful tool to differentiate symptoms of influenza from those of COVID-19 and willingness to receive COVID-19 vaccination were significantly associated with intention to receive influenza vaccination. This has been confirmed by the results of the model that investigated the determinants of the positive change in the willingness to take up influenza vaccination in those who had not undergone it in the previous six years that showed that only the variables related to the COVID-19 pandemic and not being concerned about influenza vaccine side effects were associated with this change, with females being significantly more willing.

All taken together, these findings indicate that this changed positive attitude towards influenza vaccination is related to increased trust in the effectiveness and safety of the vaccine, an increased sense of responsibility towards patients and awareness of the crucial role of influenza vaccination in the context of the COVID-19 pandemic to reduce the burden of the disease and the pressure on healthcare systems, to facilitate differential diagnosis among overlapping clinical symptoms, and to decrease the probability of severe disease as the result of coinfection with influenza and SARS-CoV-2 viruses. Therefore, according to the results of this study, there seems to be no need for educational and promotional campaigns oriented to attitudinal barriers, and also mandatory policies do not appear to represent an appropriate strategy in this context. Interventions aimed at increasing HCWs’ adherence to influenza vaccination in future seasons should probably be focused on enabling factors, such as those favoring access to and availability of vaccines. Indeed, it should also be acknowledged that campaigns involving education or promotion alone have sometimes resulted in minimal changes in vaccination rates [37,38,39,40], whereas a recent study has demonstrated the extraordinary role of enabling factors, such as the availability of worksite vaccination-dedicated clinics, in the promotion of HCWs’ adherence to recommended vaccinations [41].

### Limitations

We are aware that the study has some potential limitations that should be underlined and considered when interpreting the results. First, as with most similar research on this topic, our survey was performed as cross-sectional, and it is well known that the cross-sectional design does not allow any cause-effect relationship and poses many problems in relation to hypothesis testing since data on risk factors and outcomes are assessed at the same time. However, it was not our aim to draw conclusions on predictive relationships, but only to have insight on associations between several characteristics of HCWs and their willingness to receive influenza vaccination in the context of the COVID-19 pandemic. Therefore, the study design and setting were suitable to achieve the objectives of the study. Second, self-reported behaviors can result in the overestimation of “desirable” responses, with participants possibly having inflated compliance with recommendations of the influenza vaccination. Moreover, the information about their vaccination status was also self-reported and not based on vaccination records. This might be prone to recall, declaration, or desirability biases; therefore, an over or underestimation of coverage could have occurred. However, as the survey was self-administered and anonymous, and staff had voluntarily participated in the study, we believe that the responses were likely to be accurate with minimal social desirability bias. Previous published studies have reported a strong agreement between self-reported influenza vaccination status and the uptake documented in medical records. Therefore, this limitation is unlikely and may be weighted as minimal [42,43,44]. Finally, our study was conducted in a defined area of Southern Italy and might not be generalizable to other regions of the country. Despite these potential limitations, the extensive response to the questionnaire reduces the risk of the non-representativeness of the sample compared to the entire population.

## 5. Conclusions

The present study showed, in the context of the COVID-19 pandemic, a relevant increase in the willingness to undergo influenza vaccination, with almost all HCWs who had undergone influenza vaccination, and almost half of those who reported themselves to have never undergone influenza vaccination in previous influenza seasons, being willing to undergo it in the present season. In light of this positive attitude, interventions aimed at increasing HCWs’ adherence to influenza vaccination in future seasons should probably be focused primarily on enabling factors, such as those favoring accesses to and availability of vaccines.

## Figures and Tables

**Table 1 vaccines-09-00766-t001:** Willingness to receive influenza vaccination during the 2020/21 season according to several characteristics.

Characteristics	Total		Willingness to Receive Influenza Vaccination	
Socio-demographic and professional characteristics	N	%	N	%
Gender				
Male	221	45.6	165	74.7
Female	264	54.4	165	62.5
			χ^2^ = 8.18, 1 df, *p =* 0.004	
Age (years)	50.7 ± 10.5 *		50.9 ± 11 *	
			*t* test = −0.65, df = 431, *p* = 0.52	
Marital status				
Unmarried/widowed/separated/divorced	123	25.2	81	65.8
Married/cohabitant	365	74.8	252	69
			χ^2^ = 0.43, 1 df, *p* = 0.511	
Professional role				
Physicians	134	27.4	113	84.3
Nurses	255	52	157	61.6
Other (nursing assistants, technicians, etc.)	101	20.6	63	62.4
			χ^2^ = 22.72, 2 df, *p* < 0.001	
Current working area				
Medical	167	34.1	107	64.1
Surgical	161	32.9	109	67.7
Critical care	90	18.4	64	71.1
Laboratory and Diagnostic	72	14.6	53	73.6
			χ^2^ = 2.63, 3 df, *p* = 0.452	
**Attitudes towards influenza and COVID-19**				
Influenza is a serious disease				
Uncertain/disagree	310	63.3	184	59.3
Agree	180	36.7	149	82.7
			χ^2^ = 28.69, 1 df, *p* < 0.001	
Influenza is a preventable disease				
Uncertain/disagree	165	33.7	89	53.9
Agree	325	66.3	244	75.1
			χ^2^ = 22.45, 1 df, *p* < 0.001	
HCWs can be a source of influenza infection for patients				
Uncertain/disagree	93	19	53	57
Agree	197	81	280	70.5
			χ^2^ = 6.34, 1 df, *p* = 0.012	
Positive attitude towards influenza vaccination in HCWs				
Uncertain/disagree	117	23.9	37	31.6
Agree	373	76.1	296	79.4
			χ^2^ = 93.19, 1 df, *p* < 0.001	
Positive attitude towards vaccinations				
Uncertain/disagree	108	22	42	38.9
Agree	382	78	291	76.2
			χ^2^ = 53.76, 1 df, *p* < 0.001	
HCWs have a higher risk of getting influenza				
Uncertain/disagree	151	30.8	76	50.3
Agree	339	69.2	257	75.8
			χ^2^ = 31.15, 1 df, *p* < 0.001	
Influenza vaccination should be mandatory for HCWs				
Uncertain/disagree	326	66.5	194	59.5
Agree	164	33.5	139	84.8
			χ^2^ = 31.94, 1 df, *p* < 0.001	
The influenza vaccination is useful to distinguish influenza symptoms from COVID-19 symptoms				
Uncertain/disagree	232	47.4	123	53
Agree	258	52.6	210	81.4
			χ^2^ = 45.18, 1 df, *p* < 0.001	
Willingness to receive COVID-19 vaccination				
No	118	24.1	41	34.7
Yes	372	75.9	292	78.5
			χ^2^ = 78.74, 1 df, *p* < 0.001	
**Concerns**	**Total**		**Willingness to receive influenza vaccination**	
			**No**	**Yes**
Concern about contracting influenza in the workplace (1–10)	6.09 ± 2.62 *		5.55 ± 2.63 *	6.35 ± 2.57 *
			*t* test = −3.21, df = 488, *p* = 0.001	
Concern about the possibility to transmit influenza to hospitalized patients (1–10)	5.9 ± 2.58 *		5.01 ± 2.64 *	6.32 ± 2.45 *
			*t* test = −5.39, df = 485, *p* < 0.001	
Belief that influenza vaccine is useful to prevent influenza in hospital settings (1–10)	7.31 ± 2.32 *		5.92 ± 2.40 *	7.96 ± 1.97 *
			*t* test = −9.94, df = 487, *p* < 0.001	
Concern about influenza vaccine side effects (1–10)	5.66 ± 2.9 *		6.64 ± 2.35 *	5.20 ± 3.03 *
			*t* test = 5.25, df = 488, *p* < 0.001	
Perceived risk of contracting COVID-19 (1–10)	8.55 ± 1.73 *		8.39 ± 1.81 *	8.63 ± 1.69 *
			*t* test = −1.44, df = 485, *p* = 0.15	
Perceived risk of transmitting COVID-19 to their families (1–10)	9.17 ± 1.45 *		9.05 ± 1.51 *	9.22 ± 1.43 *
			*t* test = −1.23, df = 488, *p* = 0.22	
**Behaviors**	**N**	**%**	**N**	**%**
Influenza vaccination uptake in the previous influenza season (2019/2020)				
No	290	59.2	135	46.5
Yes	200	40.8	198	99
			χ^2^ = 149.53, 1 df, *p* < 0.001	
**Sources of information**				
Sources of information about influenza vaccination				
Scientific journals	277	56.5	210	75.8
Others (mass media, Internet, etc.)	213	43.5	123	57.8
			χ^2^ = 18.05, 1 df, *p* < 0.001	
Perceived quality of the information received on influenza vaccination				
Insufficient/low	137	28	79	56.7
Good/very good/excellent	352	72	253	71.9
			χ^2^ = 9.14, 1 df, *p* = 0.003	
Need for more information on influenza vaccination				
No	220	46.1	142	64.5
Yes	257	53.9	183	71.2
			χ^2^ = 2.42, 1 df, *p* = 0.120	

* Mean ± Standard Deviation. Number for each item may not add up to total number of study population due to missing values.

**Table 2 vaccines-09-00766-t002:** Influenza vaccination uptake in previous seasons (2014/15–2019/20) and reasons for willingness to uptake in the 2020/21 season (490 obs).

Uptake of Influenza Vaccine	N	%
Never (2014/15–2019/20)	273	55.7
2014/15	85	17.3
2015/16	89	18.2
2016/17	103	21
2017/18	132	26.9
2018/19	152	31
2019/20	200	40.8
Always (2014/15–2019/20)	75	15.3
Willingness to receive influenza vaccination in the present season (2020/21)	333	68
Willingness (2020/21) among those who had ever been vaccinated (2014/15–2019/20) (217 obs)	208	95.8
Willingness (2020/21) among those who had never been vaccinated in the previous six season (2014/15–2019/20) (273 obs)	125	45.8
Reasons for willing to uptake in the present season (2020/21) °		
Reduction of risk of infection	245	74.9
Effectiveness of the vaccine	179	55.1
Safety of the vaccine	174	53.3
Believe themselves at risk of contracting influenza	160	49.2
Reasons for not willing to uptake in the present season (2020/21) *		
Concerns about the usefulness of the vaccine	54	38.6
Considering themselves not to be at risk of contracting influenza	48	34.3
Fear of adverse events of the vaccine	36	25.7
Doubts on effectiveness of the vaccine	21	15

° Among those who were willing to receive influenza vaccination in the present season. * Among those who were not willing to receive influenza vaccination in the present season.

**Table 3 vaccines-09-00766-t003:** Multiple logistic regression analysis about willingness to receive the influenza vaccination according to several explanatory variables.

Variable	OR		95% CI	*p*
Model 1. Willingness to receive influenza vaccination				
Log likelihood = −179.88, χ^2^ = 244.91 (9 df), *p* < 0.0001, No. of obs = 481				
Willingness to receive COVID-19 vaccination	3.41		1.87–6.22	<0.001
No	1.00 *			
Yes	3.41		1.87–6.22	<0.001
Influenza vaccination uptake in the previous influenza seasons				
No	1.00 *			
Yes	2.78		1.85–4.19	<0.001
Believing that influenza vaccination is useful to distinguish influenza symptoms from COVID-19 symptoms				
No	1.00 *			
Yes	2.01		1.23–3.57	<0.001
Believing that influenza vaccine is useful to prevent influenza in hospital settings				
No	1.00 *			
Yes	1.25		1.1–1.42	0.001
Concern about the possibility to transmit influenza to hospitalized patients				
No	1.00 *			
Yes	1.15		1.03–1.29	0.011
Concern about influenza vaccine side effects				
No	1.00 *			
Yes	0.82		0.73–0.91	<0.001
Gender				
Males	1.00 *			
Females	0.99		0.58–1.67	0.959
Professional role				
Physicians	1.00 *			
Nurses	0.77		0.38–1.58	0.482
Others (nursing assistants, technicians, etc.)	0.97		0.42–2.22	0.937
Model 2. Willingness to receive influenza vaccination among those who had not received influenza vaccination in the previous six years vs. all others				
Log likelihood = −259.4, χ^2^ = 30.2 (8 df), *p* < 0.0001, No. of obs = 481				
Believing that influenza vaccination is useful to distinguish influenza symptoms from COVID-19 symptoms				
No	1.00 *			
Yes	1.65		1.03–2.63	0.035
Willingness to receive COVID-19 vaccination				
No	1.00 *			
Yes	1.65		0.92–2.95	0.094
Gender				
Males	1.00 *			
Females	1.61		1.03–2.5	0.036
Concern about influenza vaccine side effects				
No	1.00 *			
Yes	0.87		0.8–0.95	<0.001
Professional role				
Physicians	1.00 *			
Nurses	1.43		0.83–2.47	0.194
Other (nursing assistants, technicians, etc.)	1.64		0.84–3.16	0.140
Believing that influenza vaccine is useful to prevent influenza in hospital settings				
No	1.00 *			
Yes	1.04		0.93–1.16	0.426
Concern about the possibility to transmit influenza to hospitalized patients				
No	1.00 *			
Yes	1.01		0.92–1.11	0.780

* Reference category.

## Data Availability

The data presented in this study are available on request from the corresponding author.

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
