# Peer review of "Healthcare Workers’ Willingness to Receive Influenza Vaccination in the Context of the COVID-19 Pandemic: A Survey in Southern Italy"

_vaccines, 2021, doi:10.3390/vaccines9070766_

Round 1
Reviewer 1 Report
Comments to Authors
Comment: In the manuscript “Healthcare workers’ willingness to receive influenza vaccination in the context of the COVID-19 pandemic: a survey in Southern Italy”, the authors demonstrate that the willingness of the health workers (HW) for the influenza vaccination increase in the current COVID-19 pandemic in their investigation. They also describe that it is interesting for their finding to suggest the most appropriate interventions aimed at increasing seasonal influenza vaccination coverage. However, the relevance between the coverage and the COVID-19 pandemic has remained unclear. They also seem to negate their analysis by themselves with Section 4.1 Limitations. Then, the manuscript should be revised including the description about the main subject.
(Major)
- Introduction
1-1; In page 2, line 48, the cited references seem not to be adequate. At least Ref.10 describes “The coinfection did not significantly worsen the symptoms and outcomes.” in the conclusion.
- Materials and Methods
2-1; In page 2, lines 87-88, where is the data from the analysis about “beliefs about influenza and related vaccination measured on a 3-point Likert-type scale with options for agree, uncertain, and disagree” indicated?
2-2; In page 3, lines 112-113, confusing. What does “the variable“ means? Should be clarify.
- Results
3-1; In page 6, lines 147-164, the data for these descriptions could not be found. Should be clarified or also explained with another prepared Table such as Table 1.
3-2; In pages 6-7, lines 165-198, the author just list the data. Then, it is difficult to understand the meaning, significancy, relationship, their suggestions for the data, and context. Should be modified such as the ref. 8.
3-3; In Table 2, model 2, what does the “Females” means in their analysis? Is a value significancy compared with males?
- Discussion
4-1; Overall, many sentences are too long to understand the meanings, as respectively indicating in minor comments below. Then, the relevance between the coverage of influenza vaccination and the COVID-19 pandemic has remained unclear. Should re-consider the construction of all the sentences and clarify the relevance. Also, the main subject of this manuscript should be more clearly indicated.
4-2; In page 10, lines 310-328, what dose this sub-section means in the manuscript? The authors have finished just to indicate the limitations for their study. Could they confirm that the study design and setting was suitable? Should clarify it in the manuscript.
4-2; In page 10, lines 312-315, confusing. Why does the cross-sectional data cause the limitation?
(Minor)
In page 1 lines 18-23, the sentence is too long to recognize itself and the context. Should be modified briefly and clearly, but kept the meaning.
In page 8 lines 216-222, the sentence is also too long to recognize itself and the context. Should be modified briefly and clearly, but kept the meaning.
In page 8 lines 226-230, the sentence is too complicative to understand the meaning. Should be modified briefly and clearly, but kept the meaning.
In page 9 lines 247-251, the sentence is also too complicative to understand the meaning because of using many commas with participial constructions and conjunctive. Should be modified briefly and clearly, but kept the meaning.
In page 9 lines 252-255, the construction of the sentence would be complicative, because the “when” seem to be unclarity. Better to be modified.
In page 9 lines 267-270, the construction of the sentence is inappropriate, because two “although” could be found within one sentence. Should be modified.
In page 9 lines 276-281, the sentence is also too long to recognize itself and the context. Should be modified briefly and clearly, but kept the meaning.
In page 9 lines 281-287, the sentence is also too long to recognize itself and the context. Should be modified briefly and clearly, but kept the meaning.
In page 9 lines 299-304, the usage of “more….than….” in the sentence would be inappropriate. What is compared with “to educational and promotional campaigns”?
The indentations (e.g., line 40, 53, 106,114, 161, 165, 182, 256, 276,293, 316) are inappropriate. Should be modified.
Author Response
1.Introduction
1-1; In page 2, line 48, the cited references seem not to be adequate. At least Ref.10 describes “The coinfection did not significantly worsen the symptoms and outcomes.” in the conclusion.
As suggested, more pertinent references have been included.
2.Materials and Methods
2-1; In page 2, lines 87-88, where is the data from the analysis about “beliefs about influenza and related vaccination measured on a 3-point Likert-type scale with options for agree, uncertain, and disagree” indicated?
In response to this point, the statements on influenza and related vaccination were originally (in the questionnaire) presented with the 3 options for “agree”, “uncertain”, and “disagree”. However, to make reading of the results more straightforward, they were then collapsed into two categories, “agree” and “uncertain/disagree”. This strategy has now specified in the Methods section (Page 2, Lines 104-105), and the tables have been modified accordingly.
2-2; In page 3, lines 112-113, confusing. What does “the variable“ means? Should be clarify.
In response to this point we have now clarified in the text that it was an “independent variable” (Page 3, Line 138).
3.Results
3-1; In page 6, lines 147-164, the data for these descriptions could not be found. Should be clarified or also explained with another prepared Table such as Table 1.
As suggested, a new Table has been added describing the results on previous influenza vaccination (Table 2).
3-2; In pages 6-7, lines 165-198, the author just list the data. Then, it is difficult to understand the meaning, significancy, relationship, their suggestions for the data, and context. Should be modified such as the ref. 8.
In response to this point, the cited paragraph describes the results of the univariate analysis reported in Table 1. However, this was not specified in the text. We have now clarified that the data refer to Table 1 (Page 8, Lines239-241).
3-3; In Table 2, model 2, what does the “Females” means in their analysis? Is a value significancy compared with males?
In response to this point, the OR for “Females” compares the willingness of females with that of males. However, to avoid misunderstanding, we have now included for all variables the categories that are compared by ORs (now Table 3).
4.Discussion
4-1; Overall, many sentences are too long to understand the meanings, as respectively indicating in minor comments below. Then, the relevance between the coverage of influenza vaccination and the COVID-19 pandemic has remained unclear. Should re-consider the construction of all the sentences and clarify the relevance. Also, the main subject of this manuscript should be more clearly indicated.
As of your suggestion, the sentences in the Discussion have been simplified and we have more clearly stated the main subject and hypothesis of the study in the Discussion section.
4-2; In page 10, lines 310-328, what dose this sub-section means in the manuscript? The authors have finished just to indicate the limitations for their study. Could they confirm that the study design and setting was suitable? Should clarify it in the manuscript.
In response to this point, as for all published research, it is good norm to describe also the limitations that are intrinsic in all observational research. According to your suggestion, we have now clarified that, regardless of limitations, the study design and setting were suitable for the research (Page 13, Lines 401-409).
4-2; In page 10, lines 312-315, confusing. Why does the cross-sectional data cause the limitation?
In response to this point, we have now clarified in the text limitations of the cross-sectional studies. (Page 13, Lines 401-409).
(Minor)
In page 1 lines 18-23, the sentence is too long to recognize itself and the context. Should be modified briefly and clearly, but kept the meaning.
In page 8 lines 216-222, the sentence is also too long to recognize itself and the context. Should be modified briefly and clearly, but kept the meaning.
In page 8 lines 226-230, the sentence is too complicative to understand the meaning. Should be modified briefly and clearly, but kept the meaning.
In page 9 lines 247-251, the sentence is also too complicative to understand the meaning because of using many commas with participial constructions and conjunctive. Should be modified briefly and clearly, but kept the meaning.
In page 9 lines 252-255, the construction of the sentence would be complicative, because the “when” seem to be unclarity. Better to be modified.
As suggested, all of the reported sentences have been simplified in their syntax and have been shortened.
In page 9 lines 267-270, the construction of the sentence is inappropriate, because two “although” could be found within one sentence. Should be modified.
As suggested, this sentence has been modified and one of the “although” has been eliminated.
In page 9 lines 276-281, the sentence is also too long to recognize itself and the context. Should be modified briefly and clearly, but kept the meaning.
In page 9 lines 281-287, the sentence is also too long to recognize itself and the context. Should be modified briefly and clearly, but kept the meaning.
As suggested, all of the reported sentences have been simplified in their syntax and have been shortened.
In page 9 lines 299-304, the usage of “more….than….” in the sentence would be inappropriate. What is compared with “to educational and promotional campaigns”?
As suggested, this sentence has been modified so that the usage of “more….than….” is no more needed.
The indentations (e.g., line 40, 53, 106,114, 161, 165, 182, 256, 276,293, 316) are inappropriate. Should be modified.
As suggested, the indentations have all been modified.
Reviewer 2 Report
The manuscript addresses a very relevant public health topic in the field of attitudes of health care personnel in the willingness to undergo vaccinations and specifically influenza vaccination in the era of COVID-19 pandemic. The paper presents interesting results that could make a contribution for the next influenza season vaccination campaigns.
The following suggestions would improve the manuscript to be considered for publication:
- Introduction
The authors should better describe better data regarding the epidemiology of influenza in Italy, and explain how the surveillance system of influenza is set up in Italy. Moreover, the authors should include a new paragraph to describe and clarify to an international audience the policies of prevention in Italy (i.e. the National Prevention Plan and the Regional Prevention Plans) citing the appropriate literature.
2. Methods
- The survey was conducted through a self-administered questionnaire delivered to HCWs. No mention is made about reminders to increase response rate. Clarify how non-responders were managed. Please, specify if a statistical software was used to calculate the sample size.
- Specify how the independent variables to include in final multivariate logistic regression model have been selected.
- The estimated sample size was 410. Why did the authors collect 490 questionnaires?
- Please include the references to explain the 20% and 60% estimates.
- Specify whether the questionnaire was designed ad hoc for the present study.
3. Results
In Tables reporting models the standard error may be eliminated.
Please, check all the text for typos and language.
Author Response
1.Introduction
The authors should better describe better data regarding the epidemiology of influenza in Italy, and explain how the surveillance system of influenza is set up in Italy. Moreover, the authors should include a new paragraph to describe and clarify to an international audience the policies of prevention in Italy (i.e. the National Prevention Plan and the Regional Prevention Plans) citing the appropriate literature.
As suggested, the Italian surveillance system has been described, as well as the influenza prevention policies (Page 1, Lines 41-50).
- Methods
- The survey was conducted through a self-administered questionnaire delivered to HCWs. No mention is made about reminders to increase response rate. Clarify how non-responders were managed. Please, specify if a statistical software was used to calculate the sample size.
As suggested, we have now clarified how reminders were implemented, and how non-responders were managed (Page 2, Lines 86-87). No statistical software was used to calculate sample size; indeed, as mentioned in the manuscript, the single population proportion formula was used.
- Specify how the independent variables to include in final multivariate logistic regression model have been selected.
As suggested, we have specified how the independent variables were chosen for the inclusion in the models (Page 3, Lines 128-131).
- The estimated sample size was 410. Why did the authors collect 490 questionnaires?
In response to this point, we calculated the minimum sample size in 410. However, increasing the sample size may only increase precision of the estimates, so having more responses may only improve the results. We have now specified in the text that it was the “minimum” needed sample size.
- Please include the references to explain the 20% and 60% estimates.
As suggested, we have included the references explaining the 20% and 60% estimates.
- Specify whether the questionnaire was designed ad hoc for the present study.
As suggested, we have specified that the questionnaire was designed for the present study, after a thorough evaluation of the literature on the specific topic (Page 2, Lines 91-93).
- Results
In Tables reporting models the standard error may be eliminated.
As suggested, standard errors have been eliminated from the Table reporting models.
Please, check all the text for typos and language.
As suggested, the text has been revised for typos and language.
Reviewer 3 Report
The study conducted is very interesting since this is current issues worldwide.
- On line 62 - 64, what is the basis used for the assumption done.
- Has reliability test being conducted for the pilot survey involving 50 HCWs?
Overall is a great study
Author Response
- On line 62 - 64, what is the basis used for the assumption done.
As suggested, we have now included the references on which we based these assumptions.
- Has reliability test being conducted for the pilot survey involving 50 HCWs?
In response to this point, reliability was tested, and it has now been reported in the text.
Round 2
Reviewer 1 Report
Reviewer's Response-2
I’ve all checked the modified sentences in the current manuscript. The current version is also better with minor modification in the proof reading. I attached a few comments as below.
1.Introduction
1-1; In page 2, line 48, the cited references seem not to be adequate. At least Ref.10 describes “The coinfection did not significantly worsen the symptoms and outcomes.” in the conclusion.
As suggested, more pertinent references have been included.
2.Materials and Methods
2-1; In page 2, lines 87-88, where is the data from the analysis about “beliefs about influenza and related vaccination measured on a 3-point Likert-type scale with options for agree, uncertain, and disagree” indicated?
In response to this point, the statements on influenza and related vaccination were originally (in the questionnaire) presented with the 3 options for “agree”, “uncertain”, and “disagree”. However, to make reading of the results more straightforward, they were then collapsed into two categories, “agree” and “uncertain/disagree”. This strategy has now specified in the Methods section (Page 2, Lines 104-105), and the tables have been modified accordingly.
2-2; In page 3, lines 112-113, confusing. What does “the variable“ means? Should be clarify.
In response to this point we have now clarified in the text that it was an “independent variable” (Page 3, Line 138).
3.Results
3-1; In page 6, lines 147-164, the data for these descriptions could not be found. Should be clarified or also explained with another prepared Table such as Table 1.
As suggested, a new Table has been added describing the results on previous influenza vaccination (Table 2).
3-2; In pages 6-7, lines 165-198, the author just list the data. Then, it is difficult to understand the meaning, significancy, relationship, their suggestions for the data, and context. Should be modified such as the ref. 8.
In response to this point, the cited paragraph describes the results of the univariate analysis reported in Table 1. However, this was not specified in the text. We have now clarified that the data refer to Table 1 (Page 8, Lines239-241).
3-3; In Table 2, model 2, what does the “Females” means in their analysis? Is a value significancy compared with males?
In response to this point, the OR for “Females” compares the willingness of females with that of males. However, to avoid misunderstanding, we have now included for all variables the categories that are compared by ORs (now Table 3).
4.Discussion
4-1; Overall, many sentences are too long to understand the meanings, as respectively indicating in minor comments below. Then, the relevance between the coverage of influenza vaccination and the COVID-19 pandemic has remained unclear. Should re-consider the construction of all the sentences and clarify the relevance. Also, the main subject of this manuscript should be more clearly indicated.
As of your suggestion, the sentences in the Discussion have been simplified and we have more clearly stated the main subject and hypothesis of the study in the Discussion section.
4-2; In page 10, lines 310-328, what dose this sub-section means in the manuscript? The authors have finished just to indicate the limitations for their study. Could they confirm that the study design and setting was suitable? Should clarify it in the manuscript.
In response to this point, as for all published research, it is good norm to describe also the limitations that are intrinsic in all observational research. According to your suggestion, we have now clarified that, regardless of limitations, the study design and setting were suitable for the research (Page 13, Lines 401-409).
4-2; In page 10, lines 312-315, confusing. Why does the cross-sectional data cause the limitation?
In response to this point, we have now clarified in the text limitations of the cross-sectional studies. (Page 13, Lines 401-409).
R; Is it better to modify “the author” to “We” in line 386?
(Minor)
In page 1 lines 18-23, the sentence is too long to recognize itself and the context. Should be modified briefly and clearly, but kept the meaning.
In page 8 lines 216-222, the sentence is also too long to recognize itself and the context. Should be modified briefly and clearly, but kept the meaning.
In page 8 lines 226-230, the sentence is too complicative to understand the meaning. Should be modified briefly and clearly, but kept the meaning.
In page 9 lines 247-251, the sentence is also too complicative to understand the meaning because of using many commas with participial constructions and conjunctive. Should be modified briefly and clearly, but kept the meaning.
R: “, and among the predictive factors to undertake flu vaccination there was a high level of concern and perceived vulnerability to the COVID-19 pandemic [26].”; Is it better to add comma as “to undertake flu vaccination, there was〜”?
In page 9 lines 252-255, the construction of the sentence would be complicative, because the “when” seem to be unclarity. Better to be modified.
As suggested, all of the reported sentences have been simplified in their syntax and have been shortened.
In page 9 lines 267-270, the construction of the sentence is inappropriate, because two “although” could be found within one sentence. Should be modified.
As suggested, this sentence has been modified and one of the “although” has been eliminated.
In page 9 lines 276-281, the sentence is also too long to recognize itself and the context. Should be modified briefly and clearly, but kept the meaning.
In page 9 lines 281-287, the sentence is also too long to recognize itself and the context. Should be modified briefly and clearly, but kept the meaning.
As suggested, all of the reported sentences have been simplified in their syntax and have been shortened.
In page 9 lines 299-304, the usage of “more….than….” in the sentence would be inappropriate. What is compared with “to educational and promotional campaigns”?
As suggested, this sentence has been modified so that the usage of “more….than….” is no more needed.
The indentations (e.g., line 40, 53, 106,114, 161, 165, 182, 256, 276,293, 316) are inappropriate. Should be modified.
As suggested, the indentations have all been modified.
Author Response
- Is it better to modify “the author” to “We” in line 386?
As suggested, we have made the required change.
- “, and among the predictive factors to undertake flu vaccination there was a high level of concern and perceived vulnerability to the COVID-19 pandemic [26].”; Is it better to add comma as “to undertake flu vaccination, there was〜”?
As suggested, we have made the required change.